# A History of Molecular Chaperone Structures in the Protein Data Bank

**DOI:** 10.3390/ijms20246195

**Published:** 2019-12-08

**Authors:** Neil Andrew D. Bascos, Samuel J. Landry

**Affiliations:** 1Protein Structure and Immunology Laboratory, National Institute of Molecular Biology and Biotechnology University of the Philippines Diliman, Quezon City 1101, Philippines; 2Department of Biochemistry and Molecular Biology, Tulane University School of Medicine, New Orleans, LA 70112, USA; landry@tulane.edu

**Keywords:** chaperones, protein structure, protein functions, PDB

## Abstract

Thirty years ago a class of proteins was found to prevent the aggregation of Rubisco. These proteins’ ability to prevent unwanted associations led to their being called chaperones. These chaperone proteins also increased in expression as a response to heat shock, hence their label as heat shock proteins (Hsps). However, neither label encompasses the breadth of these proteins’ functional capabilities. The term “unfoldases” has been proposed, as this basic function is shared by most members of this protein family. Onto this is added specializations that allow the different family members to perform various cellular functions. This current article focuses on the resolved structural bases for these functions. It reviews the currently available molecular structures in the Protein Data Bank for several classes of Hsps (Hsp60, Hsp70, Hsp90, and Hsp104). When possible, it discusses the complete structures for these proteins, and the types of molecular machines to which they have been assigned. The structures of domains and the associated functions are discussed in order to illustrate the rationale for the proposed unfoldase function.

## 1. Introduction

The central dogma of molecular biology states that genes are transcribed into messenger RNAs, which are then translated into the proteins that carry out cellular functions. While this simple and elegant principle governs most biological systems, literature from both the distant and recent past have cited complications in the cellular environment that may disrupt the flow of genetic information [1,2].

Considering the dense population of the cytosol (average protein conc: 150 mg/mL), Finka and Goloubinoff [3] proposed an inherent need to protect nascent polypeptides from “unwanted associations” that prevent the attainment of the functional protein fold. The first of these chaperone proteins was found by Sternberg in 1973 in studies of mutations that disrupted bacteriophage λ head formation. The disruption of *groEL, groES, dnaK, dnaJ,* and *grpE* was found to have deleterious effects for the growth of the bacteriophage. These genes are bacterial homologs of Hsp60, Hsp10, Hsp70, Hsp40, and the nucleotide exchange factor for the Hsp70/Hsp40 machine. Expression of these proteins was increased with heat shock treatment, leading to their label as heat shock proteins (Hsps) [4]. The functions of these proteins were validated with the generation of recombinant versions of the proteins that performed their expected functions outside the original cellular environment [5,6]. Further experimentation revealed several types of functions for different chaperone proteins, which may be attributed to the diversity of their structures.

Current structural information divides the chaperones into five major classes based on their observed molecular weights: Hsp60, Hsp70, Hsp90, Hsp104, and the small Hsps. Aside from their differences in size, the structures of these different classes are quite divergent. The Hsp60s adopt a barrel-like Anfinsen cage structure for sequestered folding of target proteins. Hsp70s and the small Hsps, on the other hand, adopt modular “clamps” for protecting extended hydrophobic structures in their targets. Hsp90s form multidomain V-shaped structures whose scissor-like motion helps refine receptor proteins, and the Hsp104s form hexameric rings that facilitate unfolding by a ratchet-like mechanism.

A survey of the molecular structures deposited at the Protein Data Bank (PDB) (www.rcsb.org; [7]) as of August 2019 shows an uneven distribution across the five classes. The most structures are available for Hsp90 with 541 entries, while several Hsp families and their homologs had more limited entries available (e.g., small Hsps (14 entries); Hsp104 (18 entries); Hsp104 homolog, ClpA (13 structures)). The sampling of structures from different organisms is also not uniform. The majority of the structures for Hsp60 are from *E. coli*. In contrast, most entries deposited for Hsp70 and Hsp90 were from *Homo sapiens*, and the limited number of structures available for Hsp104 were mainly from *Saccharomyces cerevisiae*.

This disparity in distribution coincides with several factors: (1) Clinical significance, (2) structure complexity, and (3) ease of generation. Hsp90s, which have been associated with the refinement of receptor protein structures, currently have 541 entries. Hsp104s, on the other hand, currently have only 18 entries. Unlike crystal structures of the other chaperones, the structures of the large hexameric Hsp104 were commonly determined through cryoelectron microscopy. Developments in this field have bypassed the need to combine structural information for component functional domains to predict their interactions. For example, the Hsp70 substrate binding domain (PDBID: 1DKX) and the Hsp70 ATPase domain (PDBID: 1DKG) may be docked into hypothetical combined structures. Direct analysis of the complete chaperone proteins (e.g., complete Hsp70 structure: PDBID: 4B9Q) has provided insight into their functional mechanisms. The analysis of the available structures for each of the chaperone classes and their proposed mechanisms of action are provided in the following sections. It must be noted that the presented structures were selected based on the keyword search function at www.rcsb.org [7], choosing search results for UniProt [8] molecule name (e.g., Hsp70). This process may not cover all related structures from other homologous proteins (e.g., *H. sapiens* homolog, Hsc70). A summary of chaperone proteins curated in the UniProt database [8] is provided in Table 1. For each of the succeeding sections on chaperone families, a report is made on available component structures and how these fit unto structures of the full functional molecules. The authors make no claims on the validity of the deposited structures presented. Caveat emptor.

## 2. Review

### 2.1. Hsp60s (Chaperonins/Heptameric Anfinsen Cages)

The Hsp60 family of proteins is best known through the *E. coli* GroEL/GroES system. Mutations on these proteins have been linked with the defective maturation of bacteriophage proteins as early as 1973 [4]. Structural studies using negative stain electron microscopy revealed the involvement of a double-stacked ring made up of 60 kDA components [9]. Orthologs of GroEL are called Hsp60 and Cpn60, depending on their source, i.e., mitochondria and chloroplast, respectively [10]. These 60 kDA components combine to make stacked heptameric structures that segregate unfolded polypeptide chains from the rest of the cellular environment. Partner proteins, Hsp10 and Cpn10 (GroES in bacteria), were later discovered to regulate these Anfinsen cages, forming complete GroEL/GroES molecular machines [11].

The earliest deposited structure is from Braig et al. [12], depicting a full oligomeric structure of bacterial GroEL at 2.8 Å resolution (PDBID: 1GRL). The observed heptameric ring validated previous EM data for the macrostructure [9]. The heptameric ring was believed to form part of the Anfinsen cages where nascent polypeptides can fold. Subsequent studies have shown that the GroEL structure is able to passively promote spontaneous unfolding of stable misfolded polypeptides with exposed hydrophobic segments [1]. The open structure of the heptameric rings allows the easy dissociation of unfolded polypeptide intermediates with low binding affinity. These polypeptides may then explore alternative folding intermediates in search of more stable conformations.

The structural basis for the passive mechanism was provided by the co-crystallization of GroEL and GroES (PDBID: 1AON) [13]. This structure showed the organization of GroEL as two stacked heptamer rings, with a heptameric lid (GroES), forming a bullet-shaped GroEL–GroES_1_ structure. The subscript in GroES_1_ depicts the number of GroES molecules associated with the GroEL stack. Misfolded polypeptides may passively enter open chambers such as those formed by the “trans” (i.e., “far” from GroES) heptamer ring of GroEL. The polypeptides are then isolated from the cellular environment by the binding of a GroES lid. The Anfinsen cages formed by this association are observed in the chamber formed by GroES binding with the “cis” heptamer ring of GroEL. Conserved residues in the GroES monomers were observed to provide the regulatory function of the lid domain in this “passive” mechanism of GroEL function [14].

Interestingly, Goloubinoff et al. [1] also documented a separate function for GroEL that suggested its involvement in an active mechanism for unfolding of misfolded polypeptides. This was done in cooperation with a cochaperone protein, GroES, and was powered by ATP hydrolysis. The active mechanism involves ATP hydrolysis-dependent conformational changes that are believed to drive protein unfolding. Work by Xu et al. [13] showed different nucleotide-dependent conformations for the GroEL–GroES molecular machine. The observed bullet-shaped structure represents the heptamer rings in both the ATP-bound (closed) and ADP-bound (open) conformations. Binding interactions between the client protein and the GroEL chamber, coupled with changes in conformation induced by ATP hydrolysis, was believed to provide the mechanical forces required for client protein unfolding. That is, client proteins attached to GroEL at specific residues are pulled apart through the shift in position of these residues with nucleotide induced conformational change. In addition, the ATP bound state of GroEL promotes the binding of GroES, while the ADP-bound state prevents it. This provides another layer through which changes in bound nucleotide may provide active modulation of GroEL/Hsp60 function.

The latest published Hsp60 structures are from Nisemblat et al. [15]. This features the human mitochondrial chaperonin symmetrical “football” complex (PDBID: 4PJ1). The football-shaped form of this molecular machine was first published by Fei et al. in 2014 [16]. The structure has two GroES lids covering the two GroEL chambers. Other depictions of this molecule label it as GroEL–GroES_2_ to represent the two GroES molecules in the structure [17]. The GroEL–GroES_2_ conformation is believed to capture the protein folding functional form of the chaperones, or the predominant conformation in the population in the presence of high concentrations of target/substrate polypeptides [16].

Fei et al. [16] were able to present structures of both the substrate-bound (PDBID: 4PKN) and unbound (PBDID: 4PKO) states. A comparison between the two forms shows variance in the apical regions of GroES/Hsp10 and GroEL/Hsp60. A difference was observed in the orientation of the heptamer rings, particularly in their deviation from perfect seven-fold symmetry; this difference was associated with conformational changes that affected the flexible regions in GroEL upon GroES binding. Changes were also observed for the solvent-exposed areas inside the chamber. These changes are believed to provide the plasticity required for accommodating variably sized substrates for unfolding in the GroEL–GroES chambers.

The highest resolution attained for published Hsp60 structures is 1.7 Å; two structures are available with this resolution. The first is of a monomeric peptide fragment of *E. coli* GroEL that retains substrate binding function (PDBID: 1KID) [18]. The second 1.7 Å structure is for a homolog of the functional monomeric peptide fragment from *Thermus thermophilus* (PDBID: 1SRV) [19]. The two functional peptide fragments cover similar locations in GroEL. Both are from the *N*-terminal section (1KID: aa 184–376; 1SRV: aa 192–446) and share 69.4% identity. Solving the latter structure provided evidence for the capacity to determine high resolution structures with low acquisition times (23 min) at third generation synchrotron facilities.

The binding sites observed in these “minichaperone” structures match the expected regions for monomers in the double-stacked form. Observations made with the monomer–substrate binding show a preference for extended polypeptides in the bound structures. Its ability to capture exposed hydrophobic patches in folding intermediates may therefore force the extension of these polypeptides into the preferred bound form, providing a structural basis for the unfolding function of Hsp60 [18].

Figure 1 depicts the currently available structures of the GroEL-GroES molecular machine and its components.

### 2.2. Hsp70s (Multidomain Chaperone Systems)

In contrast to the protein folding/unfolding chambers formed by Hsp60s, the Hsp70 family of chaperones employs its actions through regulated clamping of targeted polypeptide sections. This mechanism involves the coordinated function of distinct globular domains for ATP hydrolysis and substrate binding. Work by Swain et al. [20] revealed the importance of the interdomain linker, not just for connecting the domains, but also for the regulation of their functions. Similar to the Hsp60s, the Hsp70 chaperones function in cooperation with partner proteins. Hsp40 helps both the delivery of polypeptides to the substrate binding domain, and the activation of ATP hydrolysis to promote their capture. Nucleotide exchange factors, like GrpE, facilitate the change in the Hsp70 nucleotide-binding state that promotes substrate release and resetting for succeeding target capture. The coordinated clamping action of Hsp70s has been associated with many cellular functions that involve protein unfolding, complex dissociation, and membrane translocation [21].

The multidomain nature of Hsp70 led to difficulties for the acquisition of their full structures. The earliest Hsp70-related structures in the PDB are of its substrate binding domain, with a captured polypeptide string (PDBIDs: 1DKX-1DKZ) [22]. The compact structure of the substrate binding domain allowed the investigation of its structure in crystals (PDBIDs: 1DKX-1DKZ) [22] and in solution (PDBID: 2BPR) [23]. NMR-based analysis of the latter allowed observations on the dynamic nature of the substrate binding domain both in the absence (PDBID: 1DG4) [24] and presence of a bound peptide (PDBID: 1Q5L) [25]

The structure of the missing ATPase domain was first reported in 1997 by Harrison et al. (PDBID: 1DKG) [26]. Similar to the substrate binding domain, the ATPase domain was crystalized as a distinct functional domain, separate from the rest of the Hsp70 components. This structure was of the bacterial homolog of Hsp70, DnaK. The ATPase domain was co-crystallized with an associated nucleotide exchange factor (GrpE), revealing the structural basis for this regulatory function. GrpE was observed to promote a more open conformation for the lobes that form the ATP binding-cleft, allowing faster dissociation and replacement of the attached nucleotide.

A combined structure of both ATPase and substrate binding domain was reported in 2008 (PDBID: 2V7Y) [27]. This featured both domains as undocked structures connected by an interdomain linker. The substrate binding domain for this structure lacked the helical lid, previously seen by Zhu et al. in 1996 (PDBIDs: 1DKX-1DKZ) [22]. The lid for the 2V7Y structure was truncated, and interestingly, part of the remaining C-terminal section was bound by the substrate binding cleft as a target polypeptide string. The presence of the bound polypeptide has been associated with promoting the domain-disjoined/-undocked conformation for Hsp70 [20].

Crystal structures of the substrate binding domain in the presence of long and short inhibitor peptides revealed a potential mechanism for allosteric regulation that involved the interdomain linker and the helical lid (PDBIDs: 3DPO-3DPQ) [28]. Solution NMR structures of full length (aa 1-635); and truncated (aa 1-605) versions of the *E. coli* Hsp70 homolog, DnaK, predicted locations for interaction between the different functional domains. These involved residues in the ATPase domain, the SBD, and the linker, whose signals were broadened in peptide-free / “ATP-bound” conformation (PDBID: 2KHO) [29]. While these particular structures were determined from ADP-bound molecules, comparison with previous data allowed their analysis of conformational changes between the ATP- and ADP-bound states.

An ATP-bound structure of a full Hsp70 molecule was determined in 2012 by Kityk et al. (PDBID: 4B9Q) [30]. The ATPase and substrate binding domains were observed to have a docked conformation, with the helical lid open. This facilitates substrate capture in this ATP-bound form. This open conformation in an ATP-bound Hsp70 molecule was again observed, at higher (1.9Å) resolution by Qi et al. in 2013, confirming the allosteric opening of the substrate binding domain in response to ATP binding (PDBID: 4JNE) [31]. Models of the Hsp70 domain structures in their ADP-bound and ATP-bound conformations are provided in Figure 2.

The involvement of both the bound nucleotide type and substrate binding affinity with the function of Hsp70 suggest the importance of co-chaperones/partner proteins that may facilitate these processes. The first structure of the isolated nucleotide binding domain/ATPase domain was solved in the presence of a nucleotide exchange factor, GrpE (PDBID: 1DKG) [26]. A more recent structure depicts the interaction of GrpE with the ATPase and substrate binding domains of Hsp70, as well as the interdomain linker (PDBID 4ANI) [32] (Figure 3). These interactions point to mechanisms for promoting a more open conformation for the lobes of the ATPase domain, and the stabilization of a domain-disjoined conformation in the presence of the nucleotide exchange factor. These prepare the Hsp70 molecule for ATP binding and change into the domain-docked conformation primed for substrate capture.

In addition to the nucleotide exchange factors, Hsp70 also functions in cooperation with Hsp40 cochaperones. Similar to Hsp60–Hsp10 pairing, Hsp70s pair with Hsp40s to function as a molecular machine. Hsp70s cycle through states of low and high binding affinity for their target polypeptides, as determined by their nucleotide bound state (ATP- and ADP-bound, respectively). Hsp40s regulate this process by promoting ATP hydrolysis in the nucleotide binding/ATPase domain, as well as substrate capture in the substrate binding domain. Key features of Hsp40 structure that allow these functions are its J-domain (PDBID: 1BQZ) [33], and its own substrate binding domain (PDBID: 1NLT) [34]. Hypotheses on their action suggest the promotion of substrate capture by their delivery of a target polypeptide near the substrate binding domain of the Hsp70; coincident with the promotion of ATP hydrolysis in the ATPase domain; facilitating substrate capture and retention in the ADP-bound form [35].

Genetic and biophysical studies have identified mutations that disrupt the function of Hsp40 J-domains and a conserved tripeptide sequence (HPD) between helixes 1 and 2 are important for its function [36]. There are currently no structures deposited for Hsp40 J-domain mutants in the PDB. However a recent structure shows the co-crystallization of the *E.coli* Hsp70 homolog DnaK and the N-terminally fused J-domain (PDBID: 5NRO) [30]. The combined structure allows the observation of a bond network connecting the HPD loop of the J-domain with catalytic residues in the ATPase domain of DnaK, providing a structural basis for the coordination of Hsp70 function by Hsp40 (Figure 3).

The location of the functional domains are superimposed on a domain-dissociated form of Hsp70 (white; PDBID: 2V7Y). Initial attempts at defining the molecular structure of Hsp70s were limited to the individual functional domains. The structures of the substrate binding domain (blue; PDBID: 1DKX) and the nucleotide binding domain/ATPase domain (green; PDBID: 1DKG) are shown. The combined domains were first solved with bound ADP, and they adopted a dissociated form (yellow; PDBID: 2V7Y). The helical lid of the substrate binding domain was not present in this structure. An ATP-bound form of the full structure, with a helical lid, was observed to have a conformation with the two domains docked together (red, PDBID: 4B9Q). The interdomain linker was implicated in the modulation of this domain docking, suggesting its involvement in regulating Hsp70 function.

Hsp70 binding to nucleotide exchange factors like GrpE, and its cochaperone Hsp40, regulates its function. GrpE (white) is seen to interact with the nucleotide binding domain (green) and the interdomain linker (red). This is seen to stabilize the domain-dissociated conformation in the absence of ATP (PDBID: 4ANI). In contrast, the J-domain (Jd) of Hsp40 (white) is observed to associate with the ATP-bound conformation of Hsp70 (PDBID: 5NRO). This form shows a displaced lid region for the substrate binding domain (blue). Jd is observed to interact with residues of the linker (red), which are associated with catalytic residues in the nucleotide binding domain (green). These linked interactions suggest an H-bond-based mechanism through which Hsp40 J-domains facilitate ATP hydrolysis in Hsp70 chaperones. Current analysis shows the importance of maintaining a hydrogen bond network between the HPD motif of Hsp40 and the catalytic sites of the ATPase domain to allow Hsp40-mediated stimulation of Hsp70 functions.

### 2.3. Hsp90s (Receptor Protein Refinement)

Unlike the Hsp60 and Hsp70 chaperones that serve mainly to facilitate protein folding, the Hsp90 chaperones are involved with the refinement of receptor protein structures into their functional forms. The Hsp90 chaperones have been associated with the modification of kinases, steroid hormone receptors, and transcription factors, highlighting their roles in signal transduction and gene regulation [37].

Hsp90s work through the coordinated rearrangement of three main functional domains. These proteins exist as homodimers that have N-terminal nucleotide binding domains, middle domains, and C-terminal dimerization domains. Hsp90 homodimers adopt an “open” V-structure in the ADP-bound and nucleotide-free states. Binding of ATP causes gross shifts in conformation, leading to the closing of the arms in a pincer-like motion. ATP hydrolysis reverses the change, allowing a nucleotide-dependent process for the mechanical manipulation of target protein structures. In conjunction with the effect of the bound nucleotide, interactions with several co-chaperone types (e.g., p23/Sba1; Hop/Sti1) provide additional layers of control for Hsp90 function.

Similar to the previous chaperones discussed, the modular nature of Hsp90 structures led to the separate elucidation of their functional domains, prior to the definition of their full structures. The earliest deposited Hsp90 structure in the PDB was of its nucleotide-binding domain (PDBID: 1AH8) [38]. It took another 6 years for the structure of the middle domain to be solved (PDBID: 1HK7) [39]. Lastly, the structure for the C-terminal dimerization domain was determined in 2004 (PDBID: 1SF8) [40]. While the changes observed in the individual components provide important insight on the reactions involved for Hsp90 function (i.e., structural features of ADP binding in the NBD; Huai et al. PDBID: 1Y4S [41]), the coordinated action of the combined domains is best studied in a full structure. Two crystal structures of the full Hsp90 molecule were solved in 2006. One was solved for the ADP-bound, open conformation (PDBID: 2IOQ [42]), and the other was solved for an ATP-analogue containing “closed” state (PDBID: 2CG9 [43]). The latter was stabilized by a co-crystallized co-chaperone, Sba1. Figure 4 depicts the open and closed conformations of Hsp90.

Stabilization of the closed conformation by Sba1 highlights the role of partner proteins in regulating Hsp90 functions. Several structures have been determined showing these interactions. Certain partner proteins, like Sba1, stabilize the ATP-bound form, retaining a closed conformation. In contrast, other partners (e.g., Sti1) promote ATP hydrolysis, and drive a shift towards open conformation. Interestingly, some partner proteins (e.g., Hop) also serve as scaffolds that allow the coordinated action of other chaperones (e.g., Hsp70) with Hsp90.

Hsp90s refine the structures of several protein targets: Steroid hormone receptors, protein kinases, and *Nucleotide-binding site, and leucine-rich repeat* (NLR) domain-containing proteins. Its involvement with the modification of these protein types makes it a major player in the regulation of signaling pathways, and the cellular responses to stress and immunogenic threats. In addition, Hsp90 processes RNA targets, such as small nucleolar ribonucleoproteins and RNA polymerase [37]. This further demonstrates the importance of Hsp90s for the regulation of cellular processes, with the latter functions showing its influence on gene expression.

Steroid hormone receptor processing involves three stages of structural interactions. The early stage involves the binding of the target polypeptide by the Hsp70–Hsp40 chaperone complex; the intermediate stage is achieved upon the binding of Hsp70–Hsp40 by Hsp90 through the Hop adapter protein, facilitating the transfer of the target polypeptide. Peptidylprolyl isomerase (PPI) and p23 combine with the intermediate complex prior ATP hydrolysis. Cleavage of ATP results in an Hsp90 conformational change, closing the arms, and promoting interactions between the attached cochaperones and the target polypeptide. Release of the modified polypeptide, PPI, p23, and ADP from this late complex resets the Hsp90 cycle.

Protein kinases are modified through a similar mechanism by Hsp90 chaperones; although, with different adapter molecules. The early complex also involves Hsp70 and Hsp40. This complex is bound by Hsp90 with Hop and Cdc37 (PDBID:1US7) to form the intermediate complex. Protein phosphatase 5 (Pp5) and Aha1 action promote the release of cdc37 and incorporation of ATP. Cleavage of ATP induces the conformational change that facilitates Hsp90 closing, thus enhancing protein interactions (by proximity). Aha1 then promotes the removal of ADP, releasing the attached cochaperones, and resetting Hsp90 into the open conformation.

Several structural studies have investigated the interaction of cdc37 and Hsp90. The first involved the co-crystallization of an isolated Hsp90 nucleotide binding domain and the interacting N-terminal domain of cdc37 (PDBID:1US7 [44]). Interestingly, a fit of the Hsp90 NBD unto the full structures (PDBID: 2IOQ) causes a clash between the bound cdc37 and the middle domain (Figure 5). A recent cryoEM structure of Hsp90, cdc37, and cdk4 by Verba et al. (PDBID: 5FWK) [45] shows the bound location of the M/C domains of cdc37, but the N-terminal domain was still undefined. This highlights the need for further investigation in order to determine the mechanisms of cdc37 modulation of Hsp90 function.

NLR domain-containing proteins are important receptors for pathogen recognition, and their maturation requires their association with Hsp90 and its cochaperones Rar1 and Sgt1 [46]. The process involves an initial binding of Rar1 to Hsp90 in order to stabilize its open, ATP-bound state. This allows the binding of another Rar1 molecule, which coordinates the association of a ternary complex composed of Sgt1, an NLR domain-containing protein, and Hsp90. ATP hydrolysis, induced by a key Arginine residue in the Hsp90 middle domain, changes the conformation of Hsp90 to the closed state, and this alters proximal interactions for the bound NLR domain protein, and ultimately leads to the adoption of its mature form. Lower affinity for the ADP-bound state promotes the dissociation of the bound cochaperones, replacement of the nucleotide, and the resetting of the open conformation for Hsp90.

Lastly, Hsp90 chaperones also process RNA targets. In particular, Hsp90s facilitate the assembly of small nucleolar ribonucleoproteins (snoRNPs) and RNA polymerase II. The actual mechanism of snoRNP assembly has not been elucidated; however, the formation of a functional R2TP complex has been shown to be involved [47]. This complex is formed through the linkage of Hsp90 and Rvb1/2, linked through adapter proteins Tah1 and Pih1. The Tah1–Pih1 heterodimer initially binds Hsp90 at its middle and C-terminal domains, and stabilizes its ATP-bound conformation. Rvb1/2 binding to Pih1 and the transfer of the Tah1–Pih1 heterodimer then forms the R2TP complex. These same R2TP complexes function together with Hsp90s to facilitate RNA polymerase II assembly. The R2TP–Hsp90 complex promotes the assembly of Rpb1 in the cytoplasm and its translocation into the nucleus [48].

The capacity of Hsp90s to interact with multiple partners that regulate its function highlights its role for orchestrating the refinement of both protein- and RNA-based functional structures. Its employment of a simple mechanism involving multiple binding sites, and the regulation of conformational change through the modulation of ATP hydrolysis, demonstrates the great potential of adapting defined biophysical characteristics for many different applications.

Several adapter proteins regulate the transition of Hsp90 between its open and closed conformations. Certain cochaperones like cdc37 bind the open, ADP-bound form, while others like Sba1 bind the closed, ATP-bound form. Some adapters like Aha1 can bind both forms of Hsp90. Figure 5 shows the relative binding locations for these proteins. These are based on the superimposition of Hs90 domains that were co-crystallized with the different co-chaperones (e.g. Aha1-Hsp90 middle domain (PDBID:1USU; [39]); Cdc37-Hsp90 NBD (PDBID: 1US7); Sbd1-Hsp90 full structure (PDBID:2CG9)) unto the full structure of Hsp90. Interestingly, the superimposed Cdc37 structure (PDBID:1US7) results in clashes with the other Hsp90 arm of the open Hsp90 structure (PDBID:2IOP). This suggests a role for Cdc37 inregulating the transition between open and closed conformations for Hsp90.

### 2.4. Hsp104s (Disaggregating Complexes)

Similar to the previous chaperone families, the Hsp104 proteins have been associated with the cellular stress response and the control of protein aggregation. In particular, the expression of these 104 kDA proteins were observed to be upregulated in response to thermal [49], ethanol and sodium arsenite [50], and hydrogen peroxide stress [51]. Protein aggregation has also been associated with the presence of these stress factors, and interestingly, the increased expression of Hsp104s alleviated these symptoms [52]. These findings reveal the importance of Hsp104s, not just for the maintenance of protein structures within the cell, but also as potential therapeutic agents against aggregation-based pathologies such as Alzheimer’s disease.

Relatively few entries for Hsp104 and its homologs were determined using X-ray diffraction analysis (8/12 entries for Hsp104; 16/32 entries for ClpB). The remainder is composed of structures determined by cryo-electron microscopy. The earliest deposited structures are from ClpB, the Hsp104 homolog in bacteria (*E. coli* and *T. thermophilus*). These include structures for the ClpB nucleotide binding domain 1 deposited in 2002 (PDBID: 1JBK [53]), and a structure of the full protomer deposited in 2003 (PDBID:1QVR [54]). Structures for ClpB C-terminal domains were revisited in 2012 by Biter et al. (PDBIDs: 4FCT, 4FCV, 4FCW [55]) and the structure of the second nucleotide binding domain (NBD2) was reported in 2014 (PDBID:4LJA [56]). Structures for eukaryotic Hsp104 were acquired from *S. cerevisiae* in 2017. These were for the N-terminal domain (PDBID:5U2U [57]) and the middle domain (PDBID: 5VY9 [58]). Figure 6 presents the location of the individual functional domains using the full ClpB protomer structure as reference (PDBID: 5OFO) [59].

Several structures of multimeric Hsp104 have been determined through cryo-electron microscopy (e.g. PDBIDs: 5KNE, 5VJH). These data show that Hsp104 associates into a hexameric ring that surrounds its polypeptide target. The complex exists in two main conformations (closed and extended), whose interconversion provides a “ratchet-like” function that progressively threads the target polypeptide through the central pore [58]. This mechanical motion provides a means for the translocation of target peptides through membranes, and the disaggregation/unravelling of misfolded proteins. The transition between the two forms is mediated by coordinated action of the nuclease binding domains of adjacent Hsp104 monomers in the protein complex. In the model described by Gates et al. [58], nucleotide binding domains 1 and 2 of three members of the hexameric ring (P1, P2, and P6) alternately interact with their adjacent protomer (e.g., P6 with P1; P1 with P2) via key arginine residues and their bound nucleotide. These interactions, coupled with the shift from closed to extended conformations (Figure 7), result in the mechanism that progressively pulls the target polypeptide through the central pore.

Individually crystallized functional domains of the *E. coli* homolog of Hsp104, ClbB are shown. Nucleotide binding domains 1 and 2 are shown colored red (PDBID:1JBK) and green (PDBID:4LJA), respectively. NBD2 coincides with the position of the C-terminal domain construct reported in PDBID: 4FCT. The positions of these domains were fitted onto a ClpB protomer (PDBID: 5OFO chain E) [59] for reference. The right column shows the relative position of this protomer (colored gray) in the hexameric construct (side and top views).

The figure depicts the extended (PDBID:5KNE) and closed (PDBID: 5VJH) conformations of hexameric Hsp104 structures. A captured polypeptide string is shown (colored cyan) in the central pore of the closed form. The models shown, as with all the other figures were based on the cited PDB files and rendered using Swiss PDB Viewer 4.1.0 [60]

## 3. Conclusions

Structural analysis allows us to associate particular features of the chaperone proteins to their different functional domains. Perhaps the best application of this is for the construction of fusion proteins with combined features of different selected chaperones. Some of these chaperones have been documented to work in cooperation (e.g., Hsp70 and Hsp90; [61]). Fusion proteins with chaperone components have been employed for increased efficiency of recombinant protein production [62], but there have been no reports yet regarding the direct combination of chaperone protein parts. We can probably only say that we know how a system works when we can build its functional form from its component parts. By defining the functional features of the many different chaperone domains, we build a library of resources for the design and construction of novel molecular machines with customizable functions for our desired applications.

## Figures and Tables

**Figure 1 ijms-20-06195-f001:**
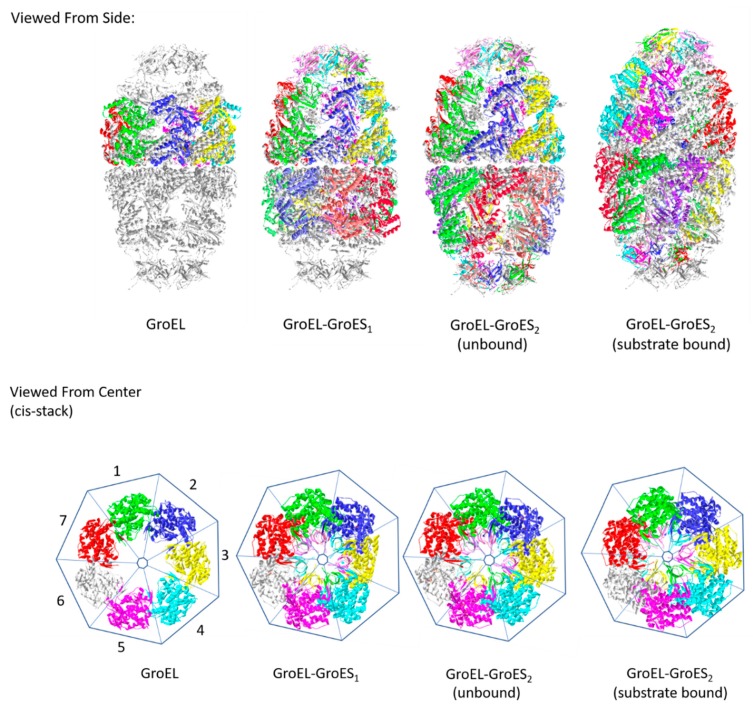
Evolution of deposited Hsp60 structures in the Protein Data Bank (PDB). Colored models depict the stated structures superimposed on the full complex structure (shown in gray). The first structure was of the GroEL-stacked heptamers (PDBID: 1GRL), followed by the structures from the co-crystallized GroES and GroEL proteins (PDBID: 1AON). This GroEL–GroES_1_ structure is also called the “bullet” conformation. The presence of another GroES forms a “football”-shaped conformation (PDBID: 4PKO). Crystallization in the presence and absence of the polypeptide substrates show a difference in the heptamer symmetry. The full structure of a human mitochondrial chaperonin (PDBID: 4PJ1) is used as a reference for the relative positions of GroEL and GroES in the available structures. Binding to a substrate is observed to alter the heptameric symmetry of the GroEL stacks. A heptagon cage is used to approximate the relative positions of the component monomers in the different states. Changes in symmetry are observed near sectors 5–7 with GroES binding. A shift in perturbed positions is seen with the binding of a polypeptide substrate (sectors 4–6).

**Figure 2 ijms-20-06195-f002:**
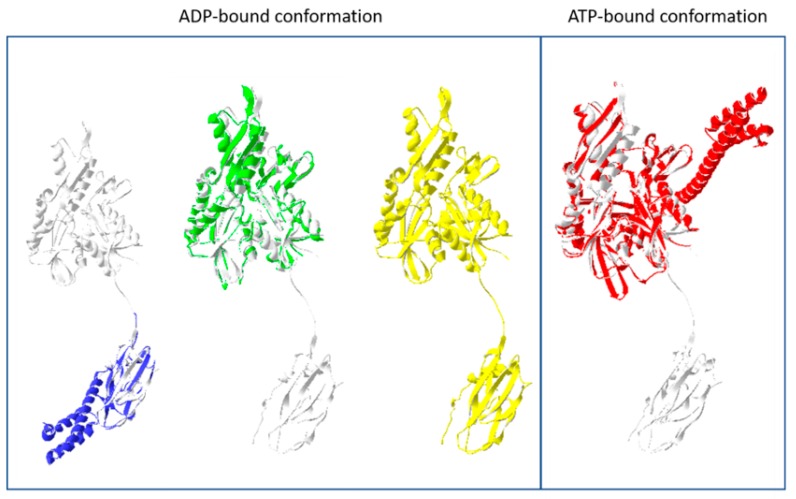
Functional domains of Hsp70 chaperones.

**Figure 3 ijms-20-06195-f003:**
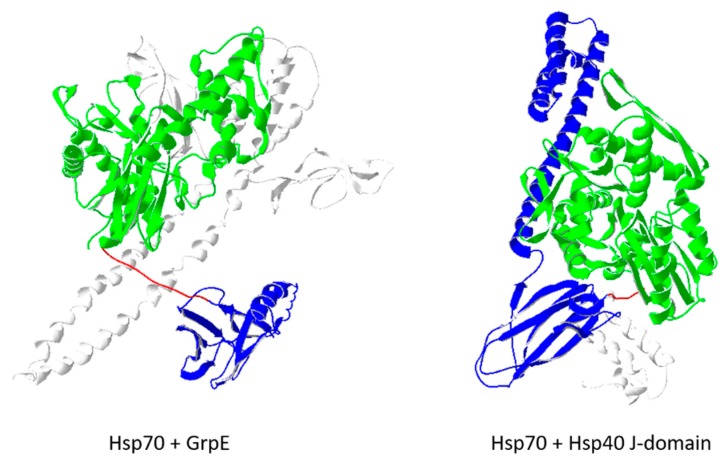
Hsp70 cochaperones.

**Figure 4 ijms-20-06195-f004:**
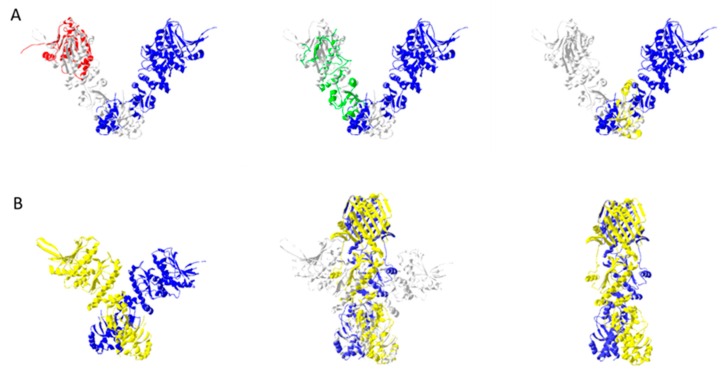
Hsp90 chaperones functional domains and conformations. Section **A** depicts the functional domains as individually crystallized, superimposed unto one arm of an open dimeric structure (PDBID: 2IOQ). The nucleotide binding domain (PDBID: 1AH8), the middle domain (PDBID: 1HK7), and the C-terminal domain (PDBID: 1SF8) are colored red, green, and yellow, respectively. Section **B** shows the nucleotide dependent transitions from an ADP-bound open form to an ATP-bound closed form. The structure of the open form was made by fitting an ADP-bound monomer structure (PDBID: 2IOQ) unto the ATP-bound dimer structure (PDBID: 2IOP), using the last two helices as bases. The monomeric arms of Hsp90 are colored yellow and blue. An overlaid structure of the open and closed conformations shows the expected movement of the arms. The structures of the open conformation are colored gray for this image.

**Figure 5 ijms-20-06195-f005:**
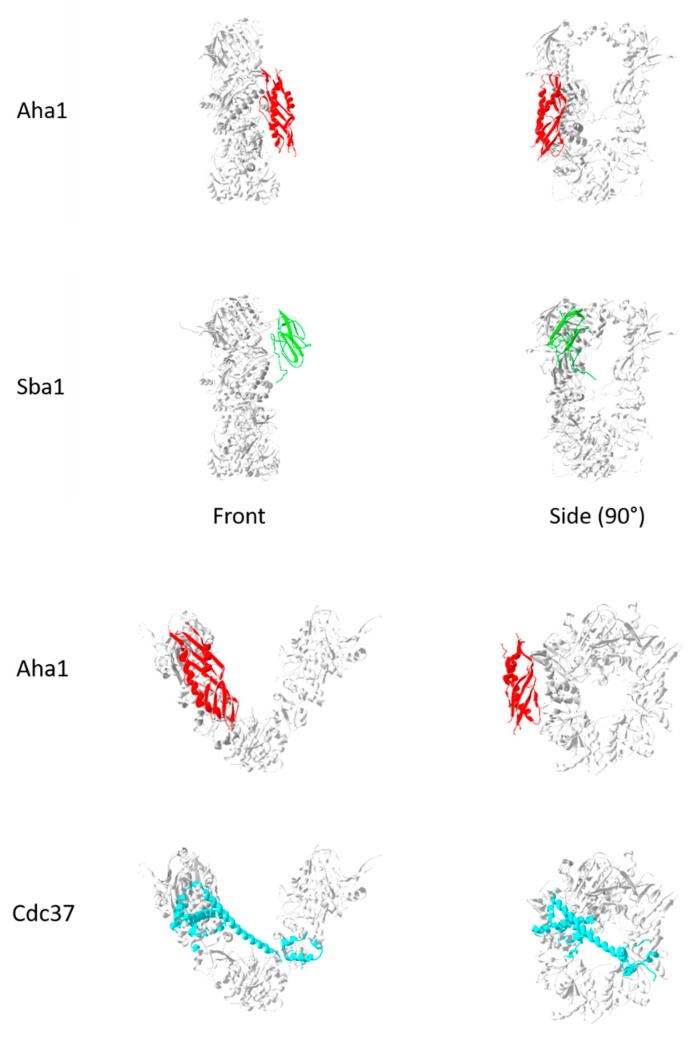
Hsp90 co-chaperones.

**Figure 6 ijms-20-06195-f006:**
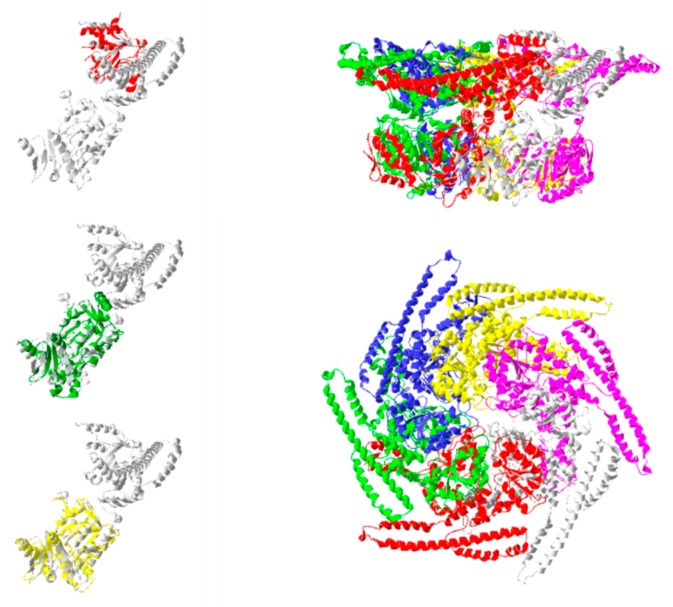
Hsp104/ClpB functional domains.

**Figure 7 ijms-20-06195-f007:**
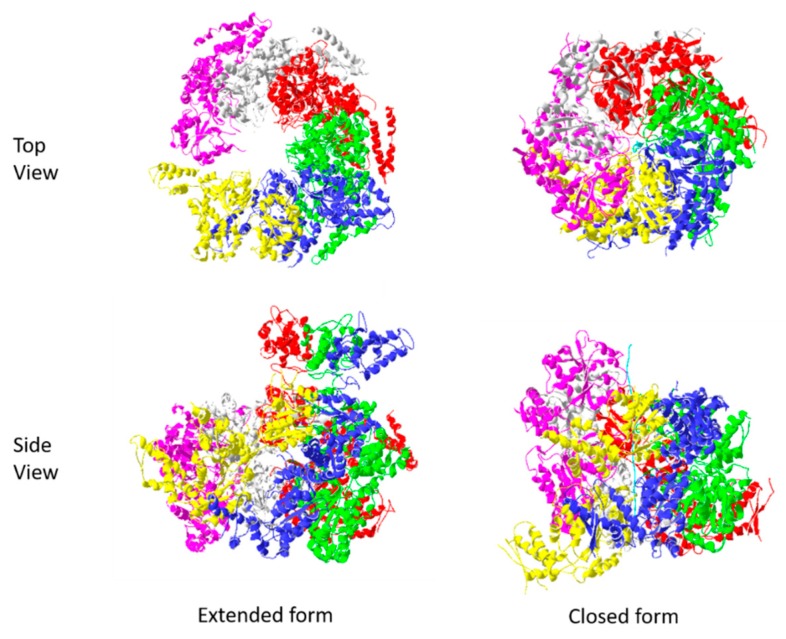
Functional conformations of Hsp104.

**Table 1 ijms-20-06195-t001:** Chaperone protein families.

Chaperone Family	Functions	Annotated Subcellular Localizations (UniProt [8]) *	Curated Samples and Related Proteins: UniProt [8] *^,++^
Hsp60	Segregate unfolded polypeptide chains Promote unfolding of misfolded polypeptides by both active and passive mechanisms	Chloroplast Cytoplasm Mitochondria	*Prokaryotic:* 60 kDa chaperonin (1–4); cpn60; groEL; groL; groL1 to groL5; mimG; prmG; thsC *Eukaryotic:* cpn60 (I and II; 1 and 2); groL(A and B); hsp60; Hspd1; Rubisco large subunit binding protein (alpha1, beta1, and beta2) Tcm62
Hsp70	Unfold misfolded polypeptides Translocate unfolded polyproteins through membranes Dissociate protein complexes	Chloroplast Cytoplasm Endoplasmic Reticulum (ER) Mitochondria Nucleus	*Viral:* Movement protein Hsp70h *Prokaryotic:* Heat shock 70 kDA protein; Hsp70; chaperone protein DnaK (1–3): [*Cochaperones:* DnaJ; GrpE (1 and 2)] HscA: [*Cochaperone:* HscB] HscC; Ssa (1 and 2); SsC1 *Eukaryotic:* Heat shock 70 kDA protein (1, 1A, 1B, 2, 3, 4, 4L, 5–10, 12–18) Chaperone Protein DnaK [*Cochaperone*: DnaJ] AtHsp70- (2,4,12,13); endoplasmic reticulum chaperone: BiP (1–5, 8); heat shock 70 kDA protein cognate (1, 2, 4, 5, II, IV); Hsc70; heat shock protein 70.2; Hsp1; hypoxia upregulated protein 1 (hyou1); Lhs1 major heat shock 70 kDA protein (Aa, Ab, Ba, Bb, Bbb, Bb); Sse (1 and 2); SsA (1–3); SsB1; SsC (1, 3); SsQ1; Sce70
Hsp90	Modification of kinases, steroid hormone receptors, and transcription factors	Cytoplasm Endoplasmic Reticulum (ER) Mitochondria Nucleus	*Prokaryotic:* Chaperone protein HtpG *Eukaryotic:* ATP-dependent molecular chaperone Hsc82; endoplasmin; endoplasmin homologs (Lpg3, Hsp90-7, grp94); heat shock cognate protein 80; heat shock-like 85 kDa protein; heat shock protein 81 (1–3); Hsp83; Hsp90 (1-6, alpha, A2, beta): [*Cochaperones:* Aha1; AHSA2P (putative); Cdc37; Cdc37-like 1; daf-41; Hch1; Hsp interacting protein, HIP; p23 (1, 2); PhPL3; peptidyl-prolyl cis trans isomerases (PPID, FKBP (4, 5, 8, 62, 65)); Protein canopy homolog 3; protein disulfide isomerase (PDI-A2, A4, A6); PPP5C; Sba1; Sgt (a, 1); OsSGT1, AtSgt1; Shu; Unc45-A,B (potential); wos2; Mod-E; Swo1
Hsp104	Dissociation, refolding, and resolubilization of protein aggregates [8]	Cytoplasm Nucleus	*Eukaryotic:* ClpB1 Hsp104

* Table 1 summarizes available data in the UniProt database for the different chaperone families discussed in the paper. Examples of these chaperones, their co-chaperones, and cellular locations are given. The results shown are from the entries curated as “Reviewed” by UniProt [8]. Other entries may have been added to the list since the manuscript was written. ^++^ Brackets denote cochaperones of the previously listed chaperone protein. Example: DnaK: [*Cochaperone*: DnaJ].

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
