# Peer review of "A History of Molecular Chaperone Structures in the Protein Data Bank"

_ijms, 2019, doi:10.3390/ijms20246195_

Round 1
Reviewer 1 Report
The Review provides a comprehensive overview on the different structural charcteristics (predominantly crytsal structures) of different HSP family members (HSP60, HSP70, HSP90, HSP104) and their co-chaperones. The structural characteristics are related to their difefrent functions.
It would be helpful if for each of the chaperone families the functions in eukaryotic and prokaryotic cells would be described a bit more detailed. Also the asepoct of the subcellular distribution of the difefrent family members would be helpful.
Otherwise the review is just a summary of all known crystal structures of the chaperones.
A Table summarizing the different memebres of the difefrent HSP families would be helpful.
The heading 2. Review is not adequate. A different title needs to be found. i.e 2. Different chaperone families.
There are a lot of typos also in names such as Golobinoff instead of Galobinoff which need to be corercted.
Author Response
|
Location |
Reviewer Comment |
Response (Action Taken) |
|
General |
The Review provides a comprehensive overview on the different structural charcteristics (predominantly crytsal structures) of different HSP family members (HSP60, HSP70, HSP90, HSP104) and their co-chaperones. The structural characteristics are related to their difefrent functions. |
Thank you for your kind words and constructive criticism. The requested changes have been incorporated in the manuscript as described below. |
|
General |
It would be helpful if for each of the chaperone families the functions in eukaryotic and prokaryotic cells would be described a bit more detailed. |
To keep the maintain the manuscript text in similar format, the additional information for functions, cellular location and examples are provided in an added table (Table 1) |
|
General |
Also the asepoct of the subcellular distribution of the difefrent family members would be helpful.
|
Please see previous response regarding Table 1 |
|
General |
A Table summarizing the different memebres of the difefrent HSP families would be helpful.
|
Table 1 was added to summarize chaperone protein data. |
|
Line 75 |
The heading 2. Review is not adequate. A different title needs to be found. i.e 2. Different chaperone families.
|
Thank you for your suggestion. The section title was named based on the prescribed format of IJMS.
With permission from the editorial board, we may replace the section heading to the suggested: 2. Different Chaperone Families |
|
General |
There are a lot of typos also in names such as Golobinoff instead of Galobinoff which need to be corercted |
Corrections made for “Galoubinoff” typographic error. (Lines 11, 33, 453). Other errors found were also corrected. |
Reviewer 2 Report
The review by Bascos and Landry focuses on the structure of molecular chaperones belonging to five protein families, and on the molecular machines they form. In general, the article is timely and can be published in Int J Mol Sci after minimal correction
Lines 11 and 33: “Goloubinoff et al.,”, not “Galoubinoff et al”. Line 38: “nuclear exchanging factor” – is it correct? It seems that nucleotide exchange factor. Line 235: “thFe ATPase domain of DnaK;” Line 293: “Nucleotide-binding site, and leucine-rich repeat (NLR)” – First bold is missing
Author Response
|
Location |
Reviewer Comment |
Response (Action Taken) |
|
General |
The review by Bascos and Landry focuses on the structure of molecular chaperones belonging to five protein families, and on the molecular machines they form. In general, the article is timely and can be published in Int J Mol Sci after minimal correction |
We thank the reviewer for his kind words and constructive criticism. The suggested changes have been incorporated in the manuscript. Other errors aside from the ones mentioned have also been corrected. |
|
Line 11 and 33: |
“Goloubinoff et al.,”, not “Galoubinoff et al.” |
The typographic errors have been corrected (Lines 11,33, 453). |
|
Line 38: |
“nuclear exchanging factor” – is it correct? It seems that nucleotide exchange factor. |
The typographic error has been corrected (Lines 38). |
|
Line 235: |
“thFe ATPase domain of DnaK;” |
The typographic error has been corrected (Lines 235). |
|
Line 293: |
“Nucleotide-binding site, and leucine-rich repeat (NLR)” – First bold is missing |
The typographic error has been corrected (Lines 293). |